# Action video game play facilitates "learning to learn"

Ru-Yuan Zhang[1,2,3,13], Adrien Chopin[4,5,6,13], Kengo Shibata[4,5], Zhong-Lin Lu [7,8,9], Susanne M. Jaeggi[10], Martin Buschkuehl[11], C. Shawn Green [12] & Daphne Bavelier[4,5 ✉]

Previous work has demonstrated that action video game training produces enhancements in a wide range of cognitive abilities. Here we evaluate a possible mechanism by which such breadth of enhancement could be attained: that action game training enhances learning rates in new tasks (i.e., "learning to learn"). In an initial controlled intervention study, we show that individuals who were trained on action video games subsequently exhibited faster learning in the two cognitive domains that we tested, perception and working memory, as compared to individuals who trained on non-action games. We further confirmed the causal effect of action video game play on learning ability in a pre-registered follow-up study that included a larger number of participants, blinding, and measurements of participant expectations. Together, this work highlights enhanced learning speed for novel tasks as a mechanism through which action video game interventions may broadly improve task performance in the cognitive domain.

[1] Institute of Psychology and Behavioral Science, Shanghai Jiao Tong University, 200030 Shanghai, China. [2] Shanghai Mental Health Center, Shanghai Jiao Tong University School of Medicine, 200030 Shanghai, China. [3] Department of Brain and Cognitive Sciences and Center for Visual Sciences, University of Rochester, Rochester, NY 14628, USA. [4] Faculté de Psychologie et Science de l'Éducation, University of Geneva, Geneva, Switzerland. [5] Campus Biotech, Geneva, Switzerland. [6] Sorbonne Université, INSERM, CNRS, Institut de la Vision, Paris, France. [7] Division of Arts and Sciences, NYU Shanghai, Shanghai, China. [8] Center for Neural Science and Department of Psychology, New York University, New York, NY 10003, USA. [9] NYU-ECNU Institute of Brain and Cognitive Science at NYU Shanghai, Shanghai, China. [10] School of Education and School of Social Sciences (Department of Cognitive Sciences), University of California, Irvine, Irvine, CA 92697, USA. [11] MIND Research Institute, Irvine, CA 92617, USA. [12] Department of Psychology, University of Wisconsin-Madison, Madison, WI 53706, USA. [13] These authors contributed equally: Ru-Yuan Zhang, Adrien Chopin. ✉email: daphne.bavelier@unige.ch

A growing body of research indicates that training on action video games enhances performance, not just on the games themselves, but in a wide range of cognitive tasks (see meta-analyses in ref. [1]). The broad generalization of skills generated by action video game play stands in contrast to the benefits induced by many conventional lab-based perceptual/cognitive training paradigms, which are often specific to trained features. For example, the benefits of perceptual training often disappear when minor changes are made to the trained stimulus, such as its orientation or position[2–4].

The mechanisms underlying how learning generalizes from trained to untrained contexts is a fundamental issue in the study of learning. Starting from the seminal studies of Thorndike at the turn of the 20th century[5], a number of distinct mechanisms that may promote broad performance improvements have been suggested. One such mechanism, termed "learning to learn"[6], is at play when information or skills gained by experience on one task (or set of tasks) permits individuals to learn new tasks faster. Such a mechanism has been of interest in a variety of learning domains, including perceptual learning and novel shape categorization[7], education[8], and machine learning[9]. It has also recently been proposed as a possible route through which action video game play produces broad generalization[10]. According to this view, action video game training produces widespread cognitive enhancements at least partially by enhancing the players' ability to learn new tasks, and, more specifically, by producing improvements in the ability to quickly extract task-relevant properties (e.g., templates for targets of interest or the timing of events).

Consistent with this framework, a few studies point to more efficient learning in habitual action video game players as compared to non-action video gamers on perceptual tasks[11–13]. However, these studies were cross-sectional (i.e., comparing individuals who choose to play action video games as part of their daily life against individuals who choose to seldom play video games). Therefore, they could not speak to whether a causal relationship exists between the act of playing action video games and faster learning. Furthermore, one of the key predictions based on the "learning to learn" framework is that its benefits should extend beyond the perceptual sub-domain. However, no work has examined whether such faster learning extends to domains other than learning in the perceptual domain. Here we directly assessed the hypothesis that action video game experience results in "learning to learn" in the context of both a "lower-level" perceptual learning task (a Gabor orientation discrimination task which we refer to below as the "orientation learning task") and a "higher-level" cognitive learning task (a dual $N$-back task which we refer to below as the "working memory learning task"). A working memory learning task was chosen partially because such tasks are known to involve a number of core constituents of executive function, such as the maintenance of information and distraction inhibition[14–16]. Because individual differences in such functions predict a host of real-world outcomes (e.g., academic and job-related success, see refs. [17,18]), and act as key behavioral markers of several psychiatric disorders[19,20], methods to improve such functions may have significant translational utility.

Here we report the results of two intervention studies conducted at two distinct geographic locations, investigating whether action video game experience improves the ability to learn new tasks. The first study was a small-scale ($N = 25$) initial intervention study, while the second was a larger-scale pre-registered replication intervention study ($N = 52$) that extended the first study by implementing a number of methodological improvements such as experimenter blinding and assessments of participant expectations.

To summarize, we show that individuals who were trained on action video games subsequently exhibited faster learning in the two cognitive domains that we tested, as compared to individuals who trained on non-action games. In the follow-up study, we confirmed that the causal effect of action video game play on learning ability is not due to participant's attention control, expectations, intrinsic motivation or flow state during the intervention.

## Results

**Study 1—initial intervention study, establishing the causal impact of action video game play on "learning to learn" across cognitive domains.** We investigated whether action video gaming facilitates "learning to learn" by selecting two representative learning tasks in cognitive science—an orientation learning task (perceptual, Fig. 1b) and a working memory learning task (cognitive, Fig. 1d). On both tasks, cross-sectional work has shown that habitual action video game players learn the tasks faster than non-video game players (orientation learning task data published in ref. [11]; cross-sectional study using the working memory learning task, presented here in Supplementary Notes 1 and 4, Supplementary Fig. S1). In order to assess whether this observed relation between action video game play and improved learning of new tasks is causal, we conducted two long-term intervention studies. In the initial intervention study, 33 participants were recruited at the University of Rochester and randomly assigned to play either a set of 3 action video games ($n = 18$) or a set of 3 control video games ($n = 15$) for 45 h (15 h per game). A total of 25 participants (14 in the action video game group and 11 in the control group) completed the initial intervention study. In order to assess the impact of training on learning abilities, all participants underwent a baseline motion learning task before training (to establish that no pre-existing differences in perceptual learning rate were seen between the randomly assigned groups), and then an orientation learning task (perceptual) and a working memory learning task (cognitive) after training (see Fig. 1). To quantify group differences on the learning tasks, we performed a hierarchical Bayesian analysis separately for each task. The learning curves for the orientation learning and the baseline motion learning tasks (Fig. 1c) were modeled as a power function of training sessions with three free parameters: initial performance, final performance, and learning rate (adapted from ref. [21]). The learning curve for the working memory learning task at post-training was modeled as a linear function with two free parameters (Fig. 1e): learning rate (slope) and initial performance (intercept).

We first confirmed that the two training groups had comparable perceptual learning performance before video game training. In the baseline motion learning task at pre-test, none of the three estimated parameters differed between the groups (learning rate, $t(23) = 0.29$, $p = 0.77$, Hedge's $g = 0.12$, $BF_{01} = 2.62$; final performance, $t(23) = 0.03$, $p = 0.98$, Hedge's $g = 0.01$, $BF_{01} = 2.704$; initial performance, $t(23) = 1.91$, $p = 0.07$, Hedge's $g = 0.77$, $BF_{01} = 0.751$). The two groups also had comparable performance in the baseline $N$-back task ($t(23) = 0.98$, $p = 0.34$, Hedge's $g = 0.41$, $BF_{01} = 1.91$; Supplementary Fig. S2).

We then tested our core hypothesis that the action-trained video game group would show faster learning than the control-trained video game group following training. Consistent with the "learning to learn" hypothesis, action video game trainees showed higher learning rates than control video game trainees in the orientation learning task (Fig. 2a, b; $t(23) = 2.16$, $p = 0.041$, Hedge's $g = 0.91$, $BF_{01} = 0.536$). Action video game trainees also performed better from the start on this task (Fig. 2c; $t(23) = 2.27$,

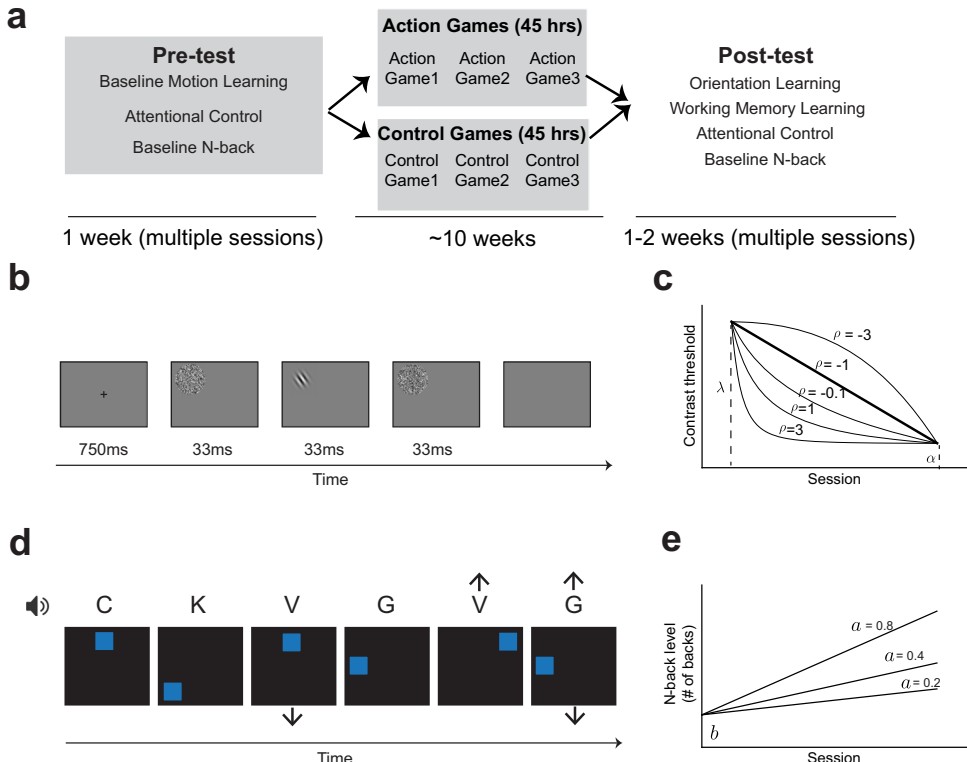

**Fig. 1 Intervention protocol, learning tasks, and learning models. a** illustrates the design of the protocol employed in both the initial (Study 1) and the replication (Study 2) intervention studies. During a pre-test, participants were assessed on a baseline motion learning task (perceptual), an attentional control task, and a baseline $N$-back task (Fig. S1A for additional task detail). Participants were then randomly assigned to one of two training groups – the action video game training group or the control video game training group. In each group, participants underwent three games each of 15 h. After a 45-h video game intervention, participants were assessed at post-test on the same attentional control task and baseline $N$-back task administered at pre-test, followed by an orientation learning task (**b**) and a working memory learning task (**d**). In the orientation learning task (**b**), participants were presented with a Gabor stimulus in one of four quadrants of the screen, and the Gabor stimulus was preceded and followed by two noise patterns. Participants pressed a button to report the direction of rotation (i.e., clockwise or counterclockwise) relative to a reference angle. In the working memory learning task (**d**), participants monitored two streams of simultaneously presented information – one auditory (letters) and one visual (blue squares) stimuli. They were asked to indicate, for each stream, whether the current stimulus matched the stimulus presented $N$ trials back in their respective sequences ($N=2$ in the provided example). Stimuli marked by an arrow indicate targets, either because of a visual or an auditory match. We modeled the learning curve in the orientation learning task (**b**) as a power function with three parameters—learning rate ($\rho$), initial performance ($\lambda$), and final performance ($\alpha$). The different curves in (**c**) illustrate the impact of different values of the learning rate parameter ($\rho$), as each curve has the same initial performance and final performance values, but different learning rates. Note that a learning rate of $-1$ corresponds to a linear progression, while values increasing from $-1$ to $+$infinity correspond to progressively steeper learning curves. The learning curve in the working memory learning task (**d**) was modeled as a linear function with two free parameters—slope ($a$) and intercept ($b$). The different curves in (**e**) have the same initial performance (i.e., intercept $b$) but different learning rates (i.e., slope $a$). "Learning to learn" predicts learning curves with steeper slopes at post-training in the action video game training group as compared to the control video game training group.

$p = 0.033$, Hedge's $g = 0.95$, $BF_{01} = 0.458$). However, there was no significant group difference in the estimated final performance, suggesting the two groups eventually reached equal performance (Fig. 2d; $t(23) = 1.95$, $p = 0.064$, Hedge's $g = 0.82$, $BF_{01} = 0.713$). The action video game group also showed a higher learning rate than the control group in the working memory learning task (Fig. 2e, f; $t(23) = 4.42$, $p = 0.0002$, Hedge's $g = 1.85$, $BF_{01} = 0.009$). Initial performance as measured by the intercept parameter was not statistically different between the two groups (Fig. 2g; $t(23) = 0.93$, $p = 0.363$, Hedge's $g = 0.39$, $BF_{01} = 1.974$).

These results are consistent with the view that action video game training facilitates "learning to learn". In other words, playing action video games improves the speed of future learning to a greater degree as compared to control games.

**Study 2—replication intervention study, larger-scale pre-registered follow-up of enhanced "learning to learn".** A number

of methodological concerns ranging from small sample sizes to lack of experimenter blinding have recently been raised in the field of cognitive training[22,23]. To address these concerns, we thus sought to replicate the findings above using a similar protocol as the initial intervention study, but with a number of methodological improvements. First, we used a larger sample—69 participants were recruited at the University of Geneva, with 52 who completed the full study (27 in the action video game group and 25 in the control group). Second, the experimenters who collected participant data at the pre- and post-tests were blinded to participants' group assignments. The participants were also blinded to the purpose of the study/intent of their assigned condition, as well as to the existence of any condition other than their own. Additionally, this replication study was pre-registered on an open science platform (https://osf.io/629yx) and the methods were executed according to the pre-registered plan, except for one aspect of the data analysis where the observed structure of the final observed dataset necessitated a divergence from the pre-

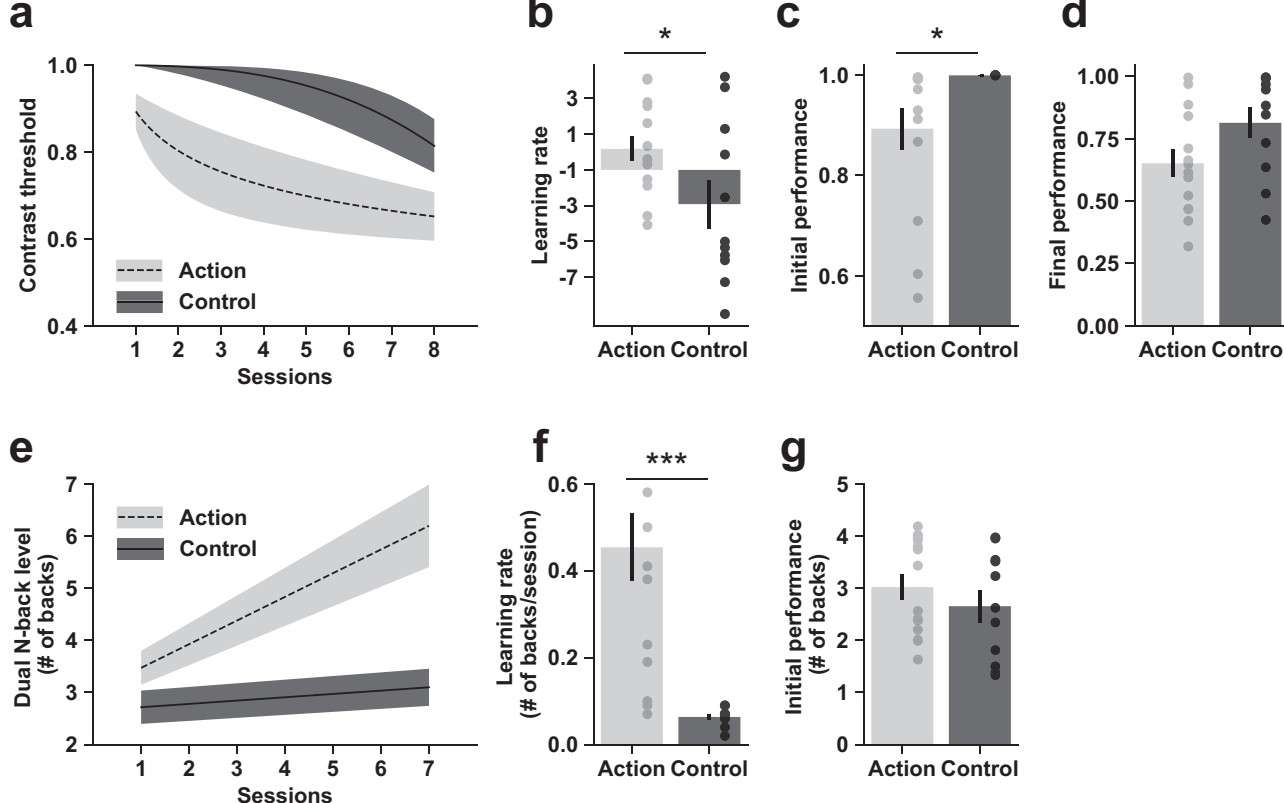

**Fig. 2 Action video game training produces "learning to learn".** (**a**) shows the impact of video game training on the orientation learning task (lower contrast thresholds represent better performance), while (**e**) shows the impact of video game training on the working memory learning task (higher $N$-back levels represent better performance). In (**a, e**), the dashed and solid lines are learning curves plotted using the group averaged estimated parameters (i.e., **b–d, f, g**). The upper and lower bounds of the shaded area are learning curves plotted using the values of the group mean ± S.E.M. The same conventions are used in Fig. 3 and Fig. S1. Estimated learning parameters in the orientation learning task (**b–d**) and the working memory learning task (**f–g**) confirm higher learning rates (**b, f**) in the action video game trainees compared with the control video game trainees. Note that the learning rates in the orientation learning task are plotted against the baseline of −1 instead of 0 because −1 indicates a linear learning progression (see Fig. 1c). Values varying from −1 to +infinity (or -infinity) indicate faster (respectively, slower) learning. All error bars are S.E.M. across participants. Significance conventions are *$p < 0.05$; **$p < 0.01$; ***$p < 0.001$. Black and gray circles correspond to each participant individual data. These conventions are kept for all figures in this paper.

registered analysis. Specifically, the pre-registered analysis was a 2(training group: action/control) × 2(time: pre-test/post-test) repeated measures ANOVA based upon the expectation that perceptual learning performance at pre-test (i.e., the baseline motion learning task) and post-test (i.e., the orientation learning task) would correlate (and thus could be considered as "repeated measures" of a common construct). Such a correlation between the two tasks was not observed (Supplementary Note 3). Given that the data violated the assumptions inherent in the pre-registered analysis, we instead opted to conduct a t-test on post-test measures only (i.e., the same analysis that was reported for the initial intervention study above; note that we nonetheless report the pre-registered analysis in Supplementary Note 3 to ensure compliance with our preregistration).

The same hierarchical Bayesian analysis used in the initial intervention study was applied to the data of the replication intervention study. We first confirmed that the two groups did not differ in the learning rates in the baseline motion learning task at pre-test ($t(49) = 0.11$, $p = 0.92$, Hedge's $g = 0.03$, $BF_{01} = 3.54$) or in terms of final performance ($t(49) = 1.0$, $p = 0.32$, Hedge's $g = 0.29$, $BF_{01} = 2.36$; Supplementary Fig. S2). The two groups, however, were not perfectly matched at pre-test, as the action video game group exhibited better initial performance than the control group (Supplementary Fig. S2; $t(49) = 2.07$, $p = 0.04$, Hedge's $g = 0.59$, $BF_{01} = 0.64$). We did not

observe significant differences in the baseline $N$-back task prior to training ($t(50) = 0.92$, $p = 0.36$, Hedge's $g = 0.26$, $BF_{01} = 2.53$).

Consistent with the results of our initial study, after training, the action video game group showed significantly faster learning, as quantified by the learning rate parameter, compared to the control group in the orientation learning task (Fig. 3a, b; $t(50) = 2.95$, $p = 0.005$, Hedge's $g = 0.83$, $BF_{01} = 0.117$). Parameter fits of initial and final performance were not significantly different across groups (Fig. 3c, d; initial performance: $t(50) = 0.03$, $p = 0.974$, Hedge's $g = 0.02$, $BF_{01} = 3.59$; final performance: $t(50) = 0.07$, $p = 0.942$, Hedge's $g = 0.01$, $BF_{01} = 3.584$). The same outcome was found in the working memory learning task, where the action video game group showed a higher learning rate than the control group (Fig. 3e, f; $t(50) = 4.06$, $p < 0.001$, Hedge's $g = 1.15$, $BF_{01} = 0.007$) with the same initial performance (Fig. 3g; $t(50) = 0.7$, $p = 0.49$, Hedge's $g = 0.2$, $BF_{01} = 2.941$).

This replication substantiates the effects of action video games in enhancing "learning to learn".

**Controlling for the role of expectations in behavioral intervention studies.** We next sought to examine the role of participant expectations and whether these could explain any of our results. Indeed, unequal game commitment or differences in expectations have been recently raised as possible confounds in

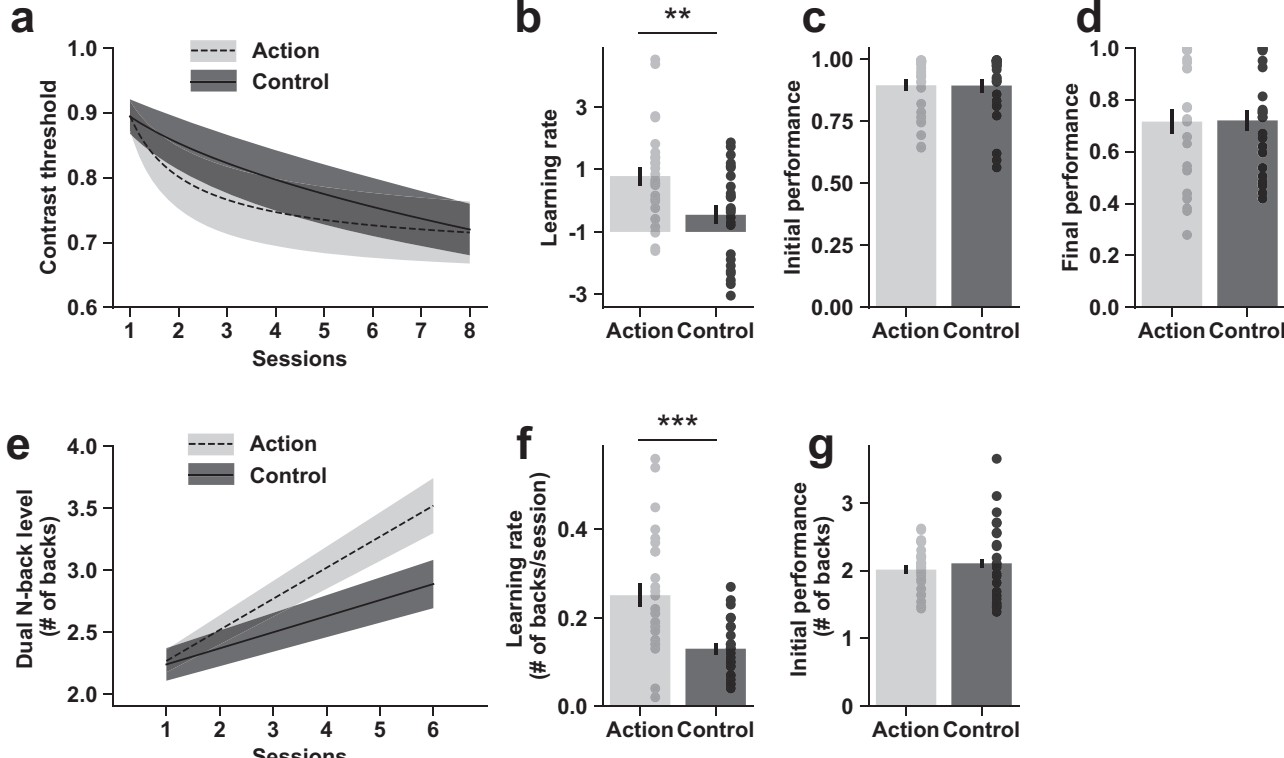

**Fig. 3 Enhanced "learning to learn" after action video game intervention in the replication intervention study.** Similar to Fig. 2, (**a**) shows the impact of video game training on the orientation learning task, while (**e**) shows the impact of video game training on the working memory learning task. In both orientation (**b**–**d**) and working memory learning (**f**, **g**) tasks, higher learning rates were observed in the action video game group compared with the control video game group (**b**, **f**). All error bars are S.E.M. across participants.

video game training studies[22,24,25]. The design of the replication intervention study controlled for experimenters' expectations but not necessarily for the participants' expectations. Indeed, although the currently recognized best practices with respect to participant blinding in behavioral experiments were employed in the replication study (i.e., participants were not made aware of either the intent of their training or the presence of another training group), it is still necessarily the case that participants were aware of their actual video game experiences. As such, it is possible that they could have formed expectations with respect to the type of games that they played[23]. To examine whether such possibilities could have had an impact on our results, we administered a debriefing questionnaire to assess participants' motivations and expectations at the completion of the initial intervention study. In the replication intervention study, a similar expectation questionnaire based on ref. [26] was administered, this time before the start of the intervention, but after having been introduced to their training game.

In the initial intervention study, we tested the subjects' belief that learning improvements, or performance on each pre- and post-test task, were induced by or related to their respective assigned games. The proportions of specific responses to the questionnaire were compared between groups. Importantly, we found no significant group differences in such expectations about the two learning tasks (Fisher's exact test, orientation learning, $p = 0.24$, $\varphi = 0.28$, working memory learning, $p = 0.21$, $\varphi = 0.33$; Supplementary Note 2). Such a result fails to support the contention that any observed differences in learning rates were caused by differences in expectation.

In the replication intervention study, we assessed participants' expectations regarding the possible effects of training with their assigned video games in a variety of domains. Of primary interest

here were expectations regarding the impact of training on their cognitive ability (the other domains assessed were mood, work productivity, and physical fitness; details in Supplementary Note 3). Participants self-reported their beliefs on the effects of the intervention on a Likert scale. Unlike in the initial study, a significant group difference was observed with regard to the expected impact on their cognitive ability. The individuals in the action group indicated a stronger expectation of improvement in cognitive ability as a result of playing their assigned game than did the individuals in the control group (t-test, $t(50) = 3.27$, $p = 0.002$, Hedge's $g = 0.91$, $BF_{01} = 0.06$). Given such a group difference in expectations, we then probed whether the expectations could potentially explain individual differences in the actual outcomes of interest by correlating expectations with learning rates at post-test. No significant correlations were found between expectations of cognition and actual learning rates (orientation learning task: $r = 0.08$, $p = 0.67$ in the action video game group, $r = -0.04$, $p = 0.85$ in the control video game group; working memory learning task: $r = -0.32$, $p = 0.10$ in the action video game group, $r = -0.06$, $p = 0.76$ in the control video game group). Finally, neither the nature of the motivation to play nor the presence of a flow state during gaming was associated with a faster learning rate at post-test (Supplementary Note 3). Taken together, our data do not provide evidence for the possibility that group differences in expectations induced the observed benefits of "learning to learn".

## Discussion
Consistent with the "learning to learn" hypothesis, we found that action video game play induces higher learning rates on novel tasks in both lower-level perceptual and higher-level cognitive domains. The effects were consistent across two separate

controlled intervention studies—an initial intervention study and a pre-registered replication intervention study. Evidence for such "learning to learn" invites a possible re-interpretation of a number of results in the literature on action video games. Specifically, the variety of cognitive domains impacted by action video game play may reflect the facilitation of learning new tasks within these domains, rather than heightened skill levels across all these domains from the outset. While the focus in much of the cognitive training literature to date has been on inducing immediate transfer, our results suggest that the capacity to learn to perform new tasks may be a useful and complementary target for future cognitive training studies.

Our work further speaks to a number of recent critiques questioning whether the positive effects of action video games observed in the literature to date are due to confounding factors, such as participant or experimenter expectations[24,27,28]. Here we followed several recommendations put forward in these critiques, such as experimenter blinding, participant blinding to conditions other than their own, and assessments of participant expectations. In particular, while participants did in some cases indicate expectations regarding the possible cognitive impact of their training conditions, these expectations were found to be unrelated to the actual learning improvements in the cognitive task used. Furthermore, we followed our pre-registered methodology, except for one analysis that diverged. This divergence highlights a key difficulty that is inherent to assessing the impact of training on learning rates in new tasks—namely that different learning tasks must be employed at pre- and post-tests to be considered novel tasks. While our expectation was that learning performance on the perceptual learning tasks at pre- and post-tests would be reasonably well correlated, this was not in fact the case. As such, although the groups were matched in terms of learning rate on the pre-test measure, it is impossible to confirm that they would have been similarly matched on the post-test measure in the absence of any training. A possible way forward to address this difficult methodological issue could be to more systematically include a no-training control, test-retest group in all training studies, as discussed in ref. [23].

The increased learning rate that we report is indicative of "learning to learn" as a consequence of action video game training; yet, one intriguing question concerns the cognitive constructs underlying this mechanism. We did not find consistent and conclusive evidence supporting the role of (i) attention control (measured by the Multiple Object Tracking task, (ii) the flow state during the intervention (measured by the Flow State Scale), or (iii) intrinsic motivation following the intervention (measured by the Intrinsic Motivation Questionnaire) in greater learning rates across studies, training groups, and/or tests (Supplementary Notes 2 and 3). Similarly, our measures of expectations suggest that possible differences in expectations concerning the training are unlikely to account for the differences in the learning rate that we report. In previous work, we have proposed that increased attention control could be one of the mechanisms through which improved learning occurs, whether for perceptual or cognitive learning[10,29]. In this view, attentional control processes, which encompass cognitive flexibility and working memory, act as a guide to identify and to keep track of task-relevant features, and thus facilitate learning[23]. This is in line with recent computational approaches to learning which also highlight the pivotal role of attention[30]. Here, however, we did not find evidence that improved learning, as measured by learning rates, was correlated with improved attention control, as measured by the Multiple Object Tracking task. While the role of attentional control on learning rate remains a promising avenue, it could also be the case that a general learning ability exists, that is involved in many unrelated tasks, including orientation and working memory

learning tasks[31]. It will be for future studies to further address these important issues.

In sum, the present work documents a pathway for cognitive training to act whereby cognitive training facilitates learning in new tasks. It also highlights the importance in future studies of considering both immediate skill performance and learning rate as potentially independent and complementary ways that cognitive enhancements may be promoted in practical applications.

## Methods

### Initial intervention study

*Participants.* 36 participants were recruited for this study, with the idea of recruiting as many as possible in an experimental timeframe between September 2014 and December 2015. Prior to being enrolled, all participants were contacted through flyers mentioning playing video games and screened for (i) video game usage; (ii) normal or corrected to normal vision; and (iii) media multitasking index (MMI). Participants were excluded if they did not have normal or corrected-to-normal vision or if they qualified as high media multitaskers (media multitasking index >5.9 as defined by ref. [29]). Three participants were excluded due to high MMI. In addition, to qualify for this initial intervention study, participants needed to have logged (1) no more than 1 h/week playing first/third-person shooter, action/action sport games or simulation games in the past year and in the year before; (2) no more than 3 h/week of play in any other video game genres in the past year; (3) no more than 5 h/week of play of any other video game genres in the year before the past year. After enrollment, participants were assigned to either the experimental (action video game) training regimen or the control (life/business simulation video game) training regimen. The assignment was done in a pseudo-random fashion so as to balance gender across training groups. Six participants failed to comply with the at-home video game training protocol; one action trainee withdrew due to game-induced motion sickness; one control trainee was excluded because of technical problems with the apparatus. The final sample thus consisted of 14 participants in the action video game group (7 women; 18–34 years old, mean age 23 years) and 11 participants (9 women; 19–56 years old, mean age 24.3 years) in the control video game group. Data from one session of the orientation learning task was missing in one action video game trainee due to a technical issue: we interpolated the missing data by duplicating the data from the preceding session. This study was run under a protocol approved by the University of Rochester Research Subjects Review Board. Informed written consent was obtained from participants during their first visit to the lab.

All participants were pre-tested in the laboratory in three 1-h sessions over 3 days, and then asked to play their assigned video games at home for a total of 45 h over a period of about 10 weeks. Finally, participants were post-tested again in the laboratory in seven 1.5-h sessions distributed over 7 days.

*Video game training and questionnaires.* Participants completed their 45 h of training by playing 15 h on each of the three assigned games administered in a randomized order (Action trainees - Call of Duty: Black Ops 1, Call of Duty: Black Ops 2, and Half-life 2; Control trainees—Sims 3, Zoo Tycoon 2013, and Viva Piñata 2006). Participants were asked to play for about 5 h per week with at least 3 h and at most 8 h per week, distributed over at least 4 different days. Gaming progress was monitored through an experimenter-assigned Microsoft Xbox account from which the participants were required to play. In addition, participants were asked to complete an online questionnaire after every play session. Participants were also asked to log the session date, starting time, ending time and a brief statement about the game experience they had. Their game play was thus monitored throughout the training period via the online game log and their Microsoft Xbox online game statistics.

In addition, participants' gaming skills were evaluated by assessing their gaming ability on assigned games at several points throughout the study. These pre- and post-training measures were obtained in the laboratory at the following time points: (1) on Day 3 of pre-test, before the 1st game training day, (2) after 15 h of training, when the participants switched from the first to the second training game, (3) after 30 h of training, when the participants switched from the second to the third training game, and (4) on Day 1 of post-test, i.e., after having completed their third training game assignment.

Given the story-based structure of the action games, the gaming performance of action video games was evaluated by tabulating checkpoints achieved during a 30-min session of a pre-selected game episode. We chose the fifth episode (S.O.G) in Call of Duty: Black Ops 1, the third episode (Old Scar) in Call of Duty: Black Ops 2, and the fourth episode (Water Hazard) in Half-life 2 as the testing regimes, based on their moderate difficulties and full coverage of necessary skills. For the control games, we initiated a completely new character in The Sims 3 and recorded how many "challenges" participants could achieve within 30 min. For Zoo Tycoon and Viva Piñata, the number of animals that participants created, purchased, and/or attracted within 30 min of play was tabulated. These game-based parameters were used to quantify the participants' gaming improvement once training was completed, and they provided an additional check that participants indeed had played their assigned games.

*Apparatus.* The attentional control task, the baseline motion learning task, and the orientation learning task were programmed in MATLAB using Psychophysics Toolbox[30,31]. The baseline *N*-back and the working memory learning tasks were programmed in E-prime 1. All pre- and post-test tasks were run under a Windows XP operating system and presented on a CRT monitor (22-inch MITSUBISHI-Diamond Pro 2070SB, 1024 × 768 resolution, 85 Hz) with linearized gamma. A video switcher was used to combine two 8-bit output channels of the graphics card so that the display system could produce gray levels with 14 bits of resolution[32]. Participants were tested in a dimly lit room, with the mean display luminance set to 58 cd/m². Monitor gamma was calibrated by fitting the best power function to the measured luminance level (Minolta Chromameter, CS-100) of 10 different gray-level settings (from 0 to 240) of the monitor (full field). Viewing was binocular at a 58 cm distance (around 2.3 arcmins per pixel) and enforced using a chin and forehead rest.

*Pre-test stimuli and procedures.* The 3 days of pre-testing consisted of a baseline motion learning task in Days 1 and 2 and then on Day 3, an attentional control task followed by a baseline *N*-back task intended as a baseline for the working memory learning task. These tasks are described in turn below.

Baseline motion learning task: We measured baseline motion learning by repeating a motion identification task that was identical to the one used in ref. [33]. The target stimulus consisted of a parafoveally presented drifting grating embedded in white 16%-RMS-contrast Gaussian image noise. Each stimulus frame lasted for 5 frames of 33 ms (165 ms) and the next stimulus always appeared 600 ms after the last response button press. The stimulus was a noisy grid (spatial frequency = 3 cycle/degree, diameter = 1.55 degrees, speed = 2.5 degrees/sec) drifting leftward or rightward. Participants indicated the direction of movement (left/right) using a keypress and received auditory feedback (high pitch if correct, low pitch if incorrect). Stimulus contrast across trials was adaptively adjusted by randomly interleaved 2/1 and 3/1 staircases (160 trials each), allowing us to derive a 75% accuracy threshold. In the first session, the initial contrast was set to 0.76 Michelson contrast for each staircase. Thereafter, for each session, the initial contrast values of the two staircases were set as the final contrast values of the two staircases from the previous session. Participants performed eight such sessions (four sessions on Day 1 and four sessions on Day 2). The dependent measure was contrast threshold.

Baseline *N*-back task: The baseline *N*-back task measures working memory ability. It was identical to the one used in ref. [34]. It consisted of a series of yellow shapes (among 8 different complex shapes) sequentially presented at the center of the screen. Each shape lasted 500 ms and was followed by a 2500 ms ISI. Participants had to indicate for each shape whether or not it matched the shape that was seen *N* trials before using key 'A' if it matched and key 'L' otherwise. Each keypress was given a neutral auditory feedback tone. Each test block consisted of 20 + *N* stimuli (i.e., trial), which included 6 target trials. Participants completed three levels of difficulty (2-back, 3-back, and 4-back) with three blocks at each *N*-back level administered in a sequential order. The dependent variable was the proportion of hits minus the proportion of false alarms averaged across all three *N*-back levels. Before the task, participants went through practice trials consisting of one block of each level of difficulty (2-back, 3-back, and 4-back) in a sequential order.

Attentional control task: The attentional control task was a Multiple Object Tracking (MOT) task using similar parameters as the ones described in ref. [35], except for a few changes listed below. Briefly, participants monitored 1 to 6 targets among a total of 16 moving stimuli. Targets were initially cued as blue sad moving faces (smileys; radius = 0.4 degree, speed = 5 degrees/s) among yellow happy moving faces for the first 2 s of a trial. Targets then turned into yellow happy faces for 4 s. Participants were asked to continue tracking the initially blue sad targets throughout a trial. The dots moved within a circular area (diameter = 20 degrees), avoiding a central area (diameter = 4 degrees). The dots followed a random trajectory, where at each frame a dot had a 60% chance of changing direction by an angle drawn from a normal distribution with a standard deviation of 12 degrees. Colliding dots and dots reaching area limits reverted directions (i.e., the dots "bounced" off one another and the aperture edge). At the end of a trial, one of the faces was cued and participants indicated by keypress whether it was among the blue sad targets cued in the beginning of the trial. A method of constant stimuli was used, with the task consisting of 65 trials in total, with 12 trials at set sizes of 2–6 targets, and 5 trials at set size of 1 target (randomized order of trial). No feedback was given, except the average score after 16 trials. All participants started with a short practice session before the measurement session. The practice session consisted of 8 trials with 2 trials at each set size from 2 to 5, presented in sequential order. The dots moved at 2 degrees per second and instant feedback was provided. The dependent measure was performance accuracy.

*Post-test stimuli and procedures.* After having completed their 45 h of video game training, participants returned, at least 48 h after the end of their training and no more than a week later, to the laboratory for a series of post-tests distributed over 7 days. On post-test Day 1, participants were administered the attentional control task followed by the baseline *N*-back task (both in the same manner as during the pre-test). The orientation learning task, as described in ref. [11], was then administered on post-test Days 2 and 3 (see details below). On post-test Day 4, one session

of a pilot task that measured the transfer of orientation learning across stimulus orientation and location were administered (this exploratory session is not reported here), followed by the first session on the working memory learning task. From Days 5 to 7, participants continued the working memory learning task, with 2 sessions per day, for a total of 7 sessions. Finally, after having completed their last working memory learning session, participants were given a debriefing expectation questionnaire.

Orientation Learning Task: The orientation learning task measures how participants learn an orientation identification task that was identical to the one in ref. [21]. A 2.1-deg-diameter circular Gabor signal temporally sandwiched between two 3-deg-diameter external noise circular patches (RMS contrast: 16%), each lasting 33 ms. The 64×64 pixel noise patch was made of individual 2×2 pixel elements. It was presented in the visual periphery (eccentricity = 5.67 degrees) at one of two locations (in the NE or SW quadrants for half the participants and in the NW or SE quadrants for the other half). Participants were presented with a 2-cpd Gabor stimuli oriented at ±12° around a reference angle of either −35° or 55° (counterbalanced across participants) and were asked to decide if the Gabor was oriented clockwise or counter-clockwise relative to the reference angle. Auditory feedback was provided after the participants' choice (high pitch noise for correct, low pitch noise for incorrect). The contrast of the Gabor stimuli across trials was adapted with high precision via two independent and randomly interleaved staircases at each of the two positions (i.e., one '1-up-2-down' 72-trials staircase and one '1-up-3-down' 84-trials staircase at both positions). During the session, signal contrast was decreased by 10% of its value after two or three successive correct responses (depending on the staircase) and increased signal contrast by 10% of its value after every error. In the first session, the initial contrast value for all staircases was set at 0.9 Michelson Contrast. For each subsequent session *N*, the initial contrast was set to the average contrast of all reversals (except the first three) from session *N*–1 (computed separately for each staircase type). The overall contrast threshold for each session was computed by averaging the thresholds across all four staircases—thereby converging to the 75% correct threshold. Each participant underwent a total of eight sessions, four sessions per day over 2 days, with 312 trials per session, with 10 additional practice trials at session 1. In addition, one single transfer session using a different Gabor orientation and different locations was carried out in the initial intervention study (but not in the replication intervention study) and will not be reported here.

Working memory learning task: The working memory learning task measures how participants learn a dual-stream *N*-back task. It duplicated parameters from the procedures of previous studies[34,36]. Participants had to perform two independent *N*-back tasks in parallel, one in the auditory modality (listening to a stream of letters) and one in the visual modality (viewing a square moving from one location to another on the screen). The letters and squares were synchronously presented at the rate of 3 s per stimulus (duration = 500 ms, ISI = 2500 ms). Participants had to indicate for each trial whether the current stimulus matched the one that was presented *N* trials back in the sequence. Participants responded with key 'A' for visual targets, and key 'L' for auditory targets; no response was required for non-targets. Participants were informed of the *N*-back level at the beginning of each block with the *N*-back level remaining fixed within a block. Each block consisted of 20 + *N* trials that included 6 targets per modality, with the first *N* trials being discarded for scoring (e.g., 22 trials for a 2-back block—the first two trials would not be counted for performance due to the absence of targets). The *N*-back level was adapted across blocks such that it increased by 1 if participants made fewer than three errors in both modalities, and decreased by 1 if they made more than five errors in either modality. In all other cases, the *N*-back level remained the same as in the previous block. A session included 15 such blocks and lasted about 25 min. Note that the first block of the first three sessions started at a 1-back level, with the following blocks changing in difficulty level according to the adaptive procedure described above. From session 4 onward, the first block of each session started at a 2-back level. The averaged *N*-back levels across the 15 blocks per session served as the dependent variable.

Debriefing Questionnaire: We developed a short questionnaire to assess the extent to which participants' expectations were related to their performance in this study. The questionnaire was administered after participants completed all experimental tasks on the last day of post-test.

### Replication intervention study

*Participants.* Our pre-registered target sample size was 50 participants. We sent invitations every month to 20 eligible participants, and we stopped offers when 64 participants had completed the pre-test part of the study, which allowed for an attrition rate of approximately 20%.

Over 300 participants were contacted through flyers mentioning playing video games, and screened for (i) video game usage; (ii) vision; and (iii) media multitasking index. Participants were not included if they did not have normal or corrected-to-normal vision as defined by binocular vision better than 20/32 on the 3m-distant SLOAN chart. They were also not included if they qualified as high media multitaskers (media multitasking index > 5.9 as defined by ref. [29]). Participants who qualified as tweeners (an intermediate profile between action video gamers and non-video gamers) were selected using the Bavelier lab video game questionnaire. Note that this questionnaire was updated mid-recruitment with an experimenter error occurring during this update, resulting in three

different questionnaires being used (22 using questionnaire #1, 3 using questionnaire #2 and 50 using questionnaire #3). The questionnaires and criteria are available on the project registration website: https://osf.io/4xe59/. All participants with psychiatric disorders, taking significant psychotropic medications or with high levels of alcohol consumption (>40 units per week) were ineligible for this study. We only selected participants who were first or second language French speakers, and those with an English comprehension self-rated at 7 out of 10 at least, as most of the experiment was conducted in French but some computer tasks required understanding English. Included participants needed to be in the age range of 18 to 35 years old.

After screening, we contacted 80 participants and 69 agreed to participate in the study. Eight participants dropped out during the study, six were excluded before training as they demonstrated no learning in the motion discrimination task (pre-training estimated learning rate of 0), another 3 had to be excluded (1 because of age outside of decided limits, 1 because they were not naive to the conditions, and 1 because they failed to comply with the procedures of the study), leaving 52 participants. The final sample thus consisted of 27 participants in the action video game group (11 women; 19–35 years old, mean age 23 years) and 25 participants (11 women; 18–33 years old, mean age 22.8 years) in the control video game group. One participant was removed from the analysis of the results in the baseline motion learning task because of a technical issue at the end of their first session; testing continued with session 2 but we could not interpolate the result of session 1. This study was run under a protocol approved by the University of Geneva Research Subjects Review Board. Informed written consent was obtained from the participants during the first visit to the lab.

Participants were assigned to either the experimental (action video game) training or the control (life/business simulation video game) training. Training group assignment was randomized using the minimization method. We applied the Efron's biased coin technique separately for each of the four following strata combining age and gender: 18–26-year-old males, 27–35-year-old males, 18–26-year-old females, 27–35-year-old females. Two independent groups of experimenters were involved in the study to ensure that the experimenters assessing performance at pre- and post-test were blinded to participant assignment. One group of experimenters (unblinded) assigned participants to training groups after pre-test, during the at-home visit (gaming material installation and start of the training). They also administered the video game training and the questionnaires. The other group of experimenters (blinded) collected all pre- and post-test outcome measures. Additionally, participants were kept naive to the existence of different training groups, and unaware of the other games that they could have been assigned to.

All participants were pre-tested in the laboratory in two 1.5-h sessions over 2 days, and then asked to play their assigned video games at home for a total of 45 h (see video game training procedure below). Finally, participants were tested again in the laboratory in two 1.5-h sessions followed by three one-hour sessions distributed over 5 days.

The recruitment, training and laboratory measurements were conducted between January 2018 and June 2018.

The complete preregistration details can be accessed via https://osf.io/629yx. This preregistration website and the project website (https://osf.io/4xe59/) also contains all administered questionnaires (e.g., video game experience questionnaire, expectation questionnaires) and participant's recruitment criteria for both intervention studies.

*Video game training and questionnaires.* We conducted the same training procedures as that in the initial intervention study, except for the following points. Unlike the initial intervention study in which questionnaires were collected after post-test, we administered several questionnaires over the course of the video game training. First, we administered expectation questionnaires based on a translated-to-French version of ref. [26] before the participants started the video game training. We asked the participants to watch the trailer videos of each of the three games that they were assigned to, and assessed with the questionnaire their expectation on how playing these games would affect their cognition, their mood, their productivity at work and their physical fitness. We were only interested in the responses about the cognition domain but probed participants in the other domains to prevent them from guessing what was our main interest. Second, we collected the Intrinsic Motivation Questionnaire (IMI questionnaire) and Flow State Scale questionnaires after a participant completed a training game (15 h of gaming), yielding three samples of the two questionnaires in total for each participant. Finally, despite informing the participants that gaming progress was monitored through a Microsoft Xbox account from which they were assigned to play, we could not access these data and monitored the self-reported logs instead.

*Apparatus.* All tasks were generated in MATLAB 2016a using the Psychophysics Toolbox and were run under a Windows 7 operating system and presented on a linearized high-performance industrial LCD glass monitor (22.5-inch ViewPixx monitor, 1920(H) × 1080(V) pixels, 120 Hz). We used the Viewpixx' M16 mode allowing to combine two 8-bit output channels of the graphics card so that the display system could produce gray levels with 14 bits of resolution[32]. Participants were tested in dimly lit light, with a mean display luminance of 58 cd/m². Monitor gamma was calibrated by fitting the best power function to the measured

luminance using a photometer at 10 different gray-scale levels. Viewing was binocular at a distance of 58 cm from the monitor, enforced using an adjustable chin and forehead rest.

*Pre-test stimuli and Procedures.* We used the same pre-test tasks as those in the initial intervention study albeit a different task arrangement. The pre-test here spanned 2 days and was run by the experimenters who were blind to group assignment. Because group assignment occurred only after pre-test, the participants were also blind to their group assignment during the pre-test. On Day 1 of pre-test, participants first completed the attentional control task and then four sessions of the baseline motion learning task. Similarly, on Day 2, participants first completed the baseline N-back task and four sessions of the baseline motion learning task. The details of the attentional control and baseline N-back tasks are documented above. In the motion learning task, signal and noise were spatially but not temporally interleaved, while in the initial intervention study or ref. [33], it was both spatially and temporally interleaved.

*Post-test stimuli and procedures.* The post-test was conducted at least 48 h after a participant completed their 45 h of gaming intervention, and at most 36 days after (mean: 7 days). We used the same tasks as those in the post-test phase of the initial intervention study but with a number of differences in the task arrangement. First, the post-test here spanned 5 days and was run by experimenters who were blind to the participants' group assignment. Second, on post-test Day 1, participants first completed the attentional control task and continued with four sessions of the orientation learning task. On Day 2, the baseline N-back task was administered and another four sessions of the orientation learning task were run. Third, on Day 3 to Day 5, participants completed 2 sessions of the working memory learning task per day, yielding a total of only six sessions of this task. Fourth, we did not run the transfer session of the orientation learning task as we did in the initial intervention study. Fifth, different from the initial intervention study, we reprogrammed the baseline N-back and working memory learning tasks using the Psychtoolbox 3.0 in Matlab R2017a.

**Data analysis, hierarchical Bayesian analysis of orientation and working memory learning tasks.** We performed a hierarchical Bayesian analysis to quantify all learning tasks. The analysis was performed separately in each group, in each task, and in each intervention study. The method of modeling for perceptual learning (orientation and baseline motion learning tasks) applies to the initial intervention study and the replication intervention study. The method of modeling for the cognitive learning (working memory learning task) applies to all three studies (i.e., the two intervention studies presented in the main text as well as the cross-sectional study presented above).

*Orientation/motion learning.* We tested the participants' ability to learn two perceptual learning tasks: orientation and motion identification. We used a power learning function to capture the decreasing trend of contrast threshold as learning proceeds because of a clear floor in the learning curves of both perceptual learning tasks (see Fig. 3 in the main text). Power learning curves have been used in the literature on perceptual learning[21]. The power function of the $i$th participant includes three parameters (1) $\rho_i$– learning rate; (2) $\lambda_i$—initial performance (at the first session); (3) $\alpha_i$ —final performance (at the last session). The threshold level ($n_{i,t}$) of the $i$th participant as a function of session ($t$) can be expressed as

$$n_{i,t} = \alpha_i + (\lambda_i - \alpha_i) * \left( \frac{t^{-\rho_i} - m^{-\rho_i}}{1 - m^{-\rho_i}} \right) \qquad (1)$$

where $m$ is the maximum session number (e.g., $m = 8$ for the initial intervention study). Three hyper distributions with six hyperparameters ($\rho_i \sim Normal(\rho_u, \rho_\sigma)$, $\lambda_i \sim Normal(\lambda_u, \lambda_\sigma)$, $\alpha_i \sim Normal(a_u, a_\sigma)$) were set accordingly to constrain the three individual learning parameters. We also gave flat priors in the range (−18, 8), (0, 13), (0, 1), (0, 4), (0, 1), (0, 4) to the six hyperparameters, respectively.

Furthermore, the psychometric function of the perceptual learning tasks was described with the commonly used Weibull function:

$$p_{i,t,k} = 1 - (1 - c) * e^{-\left( \frac{w * const_{i,t,k}}{n_{i,t}} \right)} \qquad (2)$$

where $p_{i,t,k}$ indicates the probability of the $i$th participant making a correct response in the $k$th trial of the $t$th session. $const_{i,t,k}$ is the contrast of the stimulus in this trial. $n_{i,t}$ is the $i$th participant's contrast threshold in the $t$th session. Chance level $c$ was set to 0.5. The steepness $s$ of the psychometric function was set to $2^{[37]}$. $w$ was calculated by

$$w = -\log\left( \frac{1 - \theta}{1 - c} \right)^{s^{-1}} \qquad (3)$$

where $\theta$ is the performance level that defines the threshold. In our perceptual learning tasks, $\theta$ was 0.75 from the average of one 3/1 staircase and one 2/1 staircase.

In the model, free parameters include learning parameters at both the group and the individual levels. For example, in the initial intervention study, 14 action

trainees produced 42 (3 $\rho_i$ /$\lambda_i$ /$a_i$ x 14 participants) estimated parameters at the individual level and 6 ($\rho_u$, $\rho_\sigma$, $\lambda_u$, $\lambda_\sigma$, $a_u$, $a_\sigma$) estimated parameters at the group level.

Equations 1–3 specify the complete generative model of behavioral choices when performing the orientation learning task. Leveraging the generative process, we inferred the free parameters in this hierarchical model using the Markov Chain Monte Carlo (MCMC) method implemented in the statistical package *Stan* and its python interface[38]. The hierarchical Bayesian analysis was separately applied to each group. In the fitting process, four independent Markov Chains were established, with each drawing 130,000 samples of each free parameter. The first 15,000 samples were discarded as the burn-in period, resulting in 115,000 valid samples. We found that 130,000 samples for each parameter were sufficient, as evidenced by the fact that the split R-hat statistics of all parameters for both groups were 1 (a value below 1.1 indicates a successful sampling process for a parameter, see ref. [38]). Broad uniform priors were given to the group-level hyperparameters in order to avoid bias: $\rho_u$ ~ uniform($-18$, 8), $\rho_\sigma$ ~ uniform(0, 13), $\lambda_u$ ~ uniform(0, 1), $\lambda_\sigma$ ~ uniform(0, 4), $a_u$ ~ uniform(0, 1), $_\sigma$ ~ uniform(0, 4). We also bound $\rho_i$ to ($-18$, 8), $\lambda_i$ to (0, 5) and $a_i$ to (0, 2), in order to promote the MCMC sampling efficiency. Note that all model settings were identical for the two groups in order to ensure no additional bias was introduced. Thus, any difference between groups should be attributed to the existing differences in the data.

We computed the learning parameters (i.e., $\rho_i$, $\lambda_i$, and $a_i$) of a participant by averaging the total 460,000 samples (4 chains × 115,000 valid samples) of each parameter. Statistical differences across groups on learning rate, initial, and final performance were assessed by two-sample t-tests (two-tailed) implemented in the *scipy* python package (see results in the main text).

*Working memory learning*. The working memory learning task measures how participants learn an adaptive dual N-back task. Individual participants' progress was indexed in term of N-back levels (e.g. 2-back, 3-back) which were modeled using a linear function:

$$n_{i,t} = a_i * t + b_i \qquad (4)$$

where $n_{i,t}$ is the N-back level of the ith participant in the tth session (a session is 25 blocks). Two free parameters specify the learning curve of this participant: (1) $a_i$— learning rate (the slope of the linear function); (2) $b_i$ —initial performance (the intercept of the linear function). The first three blocks of each session were removed from the analysis because each session started back to N-back level 1, and therefore no or less variability in the N-back level reached was observed during these blocks. Given the range of our data (see Fig. 2 in the main text) and our focus on detecting group difference, a linear function (only 2 degrees of freedom) appeared sufficient to capture the characteristics of learning in this task.

Furthermore, we used a hierarchical Bayesian approach assuming that all learning parameters ($a_i$ and $b_i$) of individuals follow hyper normal distributions that represent the group-level characteristics of learning: $a_i$ ~ $N(A_u, A_\sigma)$, $b_i$ ~ $N(B_u, B_\sigma)$.

Since a participant faced different N-back difficulty levels in different blocks within a session, we described the probability of a correct response in a trial using a power psychometric function:

$$p_{i,t,j,k} = g * T_{i,t,j}^{-s_{i,t}} + c \qquad (5)$$

where $p_{i,t,j,k}$ indicates the probability of the ith participant making a correct response in the kth trial of the jth block of the tth session. Similarly, $T_{i,t,j}$ is the N-back level that the participant faced in the jth block. g is the gain factor of the psychometric function. We set g to 0.45, which is equivalent to set the guessing rate to 0.05, to account for the observed guessing even in the easiest 1-back task. c is the chance level 0.5. $S_{i,t}$ is the steepness of the psychometric function in the tth session. The steepness is related to the participant's N-back threshold $n_{i,t}$ and therefore changes session by session:

$$s_{i,t} = -\frac{\log(\frac{\theta-c}{g})}{\log(n_{i,t})} \qquad (6)$$

where $\theta$ is the accuracy corresponding to the N-back threshold, which was 0.85 in this task given the adaptive stimulus setting in the experiment.

In the model, free parameters include learning parameters at both the group and the individual levels. For example, in the initial intervention study, 14 action trainees produced 28 (2 $a_i$/$b_i$ x 14 participants) free parameters at the individual level and 4 ($A_u$, $A_\sigma$, $B_u$, $B_\sigma$) free parameters at the group level. Uniform priors were given to the four hyperparameters with parameter values (0, 3), (0, 2), (0, 5), (0, 3) respectively.

Each of 4 Markov chains drew 130,000 samples, with the first 30,000 samples discarded as the burn-in period. We found 130,000 samples were sufficient for all Markov chains for this model to converge, as evidenced by the fact that the split R-hat statistics for all parameters in both groups were equal to 1.

*Other statistics and reproducibility*. For all statistics, $n = 25$ in the initial intervention study and $n = 52$ in the replication intervention study, except for the baseline motion learning task for which $n = 51$. All correlations were Spearman correlations, and statistical tests were two-tailed tests with alpha level at 5%.

**Reporting summary**. Further information on research design is available in the Nature Research Reporting Summary linked to this article.

## Data availability
All the data used to generate the statistics provided here are available at https://osf.io/4xe59.

## Code availability
All the codes used to generate the statistics provided here are available at https://osf.io/4xe59. Matlab R2017a and SPSS 12.0 were used for most analyses. All Bayes factors for two-sample comparisons were calculated using the Bayes Factor Toolbox (https://klabhub.github.io/bayesFactor/). We used the function bf.ttest2 when investigating differences between paired groups, and bf.anova with one between-subject factor when investigating differences between independent groups, because this design is equivalent to a t-test. Results were verified using the statistical software JASP 0.14 and the two tools agreed. For the hierarchical model involving the Markov Chain Monte Carlo method, we use the statistical package *Stan* and its python interface.

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

## Acknowledgements

The authors are indebted to Olga Pikul and Alyse Stegman for their invaluable help throughout these studies. We also thank Sylvie Denkinger, Marta Martins, Patricia Poma and Jérémie Todeschini for their help in data collection. We thank Long Ni and Dr. Weiji Ma for helpful discussions on computational modeling. This work was supported by an Office of Naval Research award N00014-14-1-0512 (DB&CSG) and N00014-17-1-2049 (CSG), a Swiss National Foundation Grant 100014_178814 (DB), a National Eye Institute EY020976 (DB), a Center of Visual Science training grant EY001319 (U. of Rochester Center for Visual Science), a National Eye Institute Grant EY017491 (ZLL), Natural Science Foundation of Shanghai 21ZR1434700 (RYZ), the research project of Shanghai Science and Technology Commission (20dz2260300) and the Fundamental Research Funds for the Central Universities (RYZ), National Natural Science Foundation of China 32100901 (RYZ), and a National Institute on Aging Grant 1K02AG054665 (SMJ).

## Author contributions

Research Question: D.B., C.S.G., R.Y.Z., A.C.; Study Design—orientation and baseline motion learning: Z.L.L., R.Y.Z. & D.B.; Study Design—working memory learning: S.M.J., M.B., R.Y.Z. & D.B.; Data Collection—Initial Study: R.Y.Z.; Data Collection—Replication Study: A.C., K.S.; Data Collection-cross-sectional working memory learning: S.M.J., M.B.; Data analyses: R.Y.Z., A.C., K.S. with C.S.G. & D.B.; R.Y.Z., A.C. and K.S. wrote the first draft of manuscript, D.B. and C.S.G. provided significant revision and all other authors provided comments on the manuscript.

## Competing interests

The authors declare the following competing interests: D.B. is a founding member and scientific advisor to Akili Interactive Inc, whose mission is to develop therapeutic video games. M.B. is employed at the MIND Research Institute whose interest is related to this work. S.M.J. has an indirect financial interest in the MIND Research Institute. Z.L.L. is a co-founder of and has intellectual property and personal financial interests in Adaptive Sensory Technology, Inc. (San Diego, CA). The remaining authors declare no competing interests.
