## [Transparent Peer Review File · Communications Biology]

REVIEWERS' COMMENTS:

Reviewer #1 (Remarks to the Author):

The paper by Zhang et al. examined in two studies (study 2 is also pre-registered) whether action video game facilitates "learning to learn". The design chosen by the authors is exquisite. The intervention design in both studies proves the causal impact of action video game on "learning to learn" across cognitive domains- The sample size used by the authors is valid and so is the statistical analysis supported by Bayes factors. A big issue in the field is controlling for expectations (of the experimenter and of the participants). The authors crucially controlled for experimenter blinding, participant, blinding to conditions other than their own, and assessments of participant expectations. The data presented in the two studies are consistent with the "learning to learn" hypothesis. It is difficult to find any criticisms (methodological or theoretical) to this paper and I hope that the authors will keep producing such important contributions in the field.

Reviewer #2 (Remarks to the Author):

Green and Bavelier initially reported "learning to learn" defined as enhancement of learning rates in new tasks, as a result of action game training in Nature in 2003. Ever since the initial finding, research on video game play has made tremendous influences on various fields of research including learning and memory, and education. However, recently the possibility of contamination of some artifacts in "learning to learn" resulting from video game training have been discussed. The current paper conducted experiments in which these artifacts were very carefully controlled. The results still showed clear and significant "learning to learn". The manuscript was well written. The results were robust. Above all, showing that "learning to learn" occurs by video game play training without the discussed artifacts is extremely important in the research on video game play. I do not have any particular concern. I strongly recommend this paper to be accepted for publication in Communications Biology.

Signed by Takeo Watanabe

Reviewer #3 (Remarks to the Author):

In this manuscript, Zhang et al describe a study aimed at investigating how active videogames impact the learning abilities in other non-related tasks (perceptual learning and working memory). To test this hypothesis, the authors performed two intervention studies in which separate groups of adult volunteers played either action video games or "control" video games for 45h. Perceptual learning (orientation) and working memory (n-back) were measured to assess the participants' learning abilities before and after video games play.

The main result of the study is that, compared to the control group, the action video game play group exhibited a faster learning rate both for perceptual learning and the working memory task despite initial similar learning abilities and despite reaching a comparabbble final performance.

From this result, the authors conclude that action video game play enhances the ability to "learn to learn", facilitating the ability to learn new tasks.

The experimental paradigm is sound and well controlled, I also would like to congratulate the authors for the effort in performing a second study using the pre-registration and including a large sample size. The statistical analyses are appropriate and the manuscript is overall well written. I only have a few minor points concerning the paper.

(1) I think that the figures should include single subjects's data, so that the reader could appreciate the interindividual variability of the sample. For example, the bar plot of the learning rate in the two

conditions might be either replaced or accompanied by a scatter plot.

(2) the authors controlled for several different factors that could affect the outcome of the study, including expectations, attentional abilities, motivation... however, two basic factors that might be potentially relevant have not been considered. I wonder whether the observed effect of action video games might be different across genders and with age.

(3) The discussion is very clear but a bit succinct, I think it could benefit from some speculation about the mechanisms underlying the effect of active video games on learning abilities, perhaps in the more general framework of neural plasticity.

1 Dear Editorial team,

2

3 We are really happy to hear about the acceptance in principle and thankful to the reviewers
4 for their enthusiastic response and positive feedback. Since the Reviewers 1&2 had no
5 comment to be addressed, we addressed Reviewer 3's comments below, one by one, our
6 responses appearing in blue. Changes in the text of the manuscript was highlighted in red.

7

8 Sincerely

9

10 Daphne Bavelier

11 Psychology and Education Science

12 University of Geneva, Switzerland

13 Email: daphne.bavelier@unige.ch

14

15

16

17 REVIEWERS' COMMENTS:

18

19 Reviewer #3 (Remarks to the Author):

20

21 In this manuscript, Zhang et al describe a study aimed at investigating how active videogames
22 impact the learning abilities in other non-related tasks (perceptual learning and working
23 memory). To test this hypothesis, the authors performed two intervention studies in which
24 separate groups of adult volunteers played either action video games or "control" video games
25 for 45h. Perceptual learning (orientation) and working memory (n-back) were measured to
26 assess the participants' learning abilities before and after video games play.

27 The main result of the study is that, compared to the control group, the action video game play
28 group exhibited a faster learning rate both for perceptual learning and the working memory
29 task despite initial similar learning abilities and despite reaching a comparable final
30 performance.

31 From this result, the authors conclude that action video game play enhances the ability to
32 "learn to learn", facilitating the ability to learn new tasks.

33

34 The experimental paradigm is sound and well controlled, I also would like to congratulate the
35 authors for the effort in performing a second study using the pre-registration and including a
36 large sample size. The statistical analyses are appropriate and the manuscript is overall well
37 written. I only have a few minor points concerning the paper.

38

39 (1) I think that the figures should include single subjects's data, so that the reader could
40 appreciate the interindividual variability of the sample. For example, the bar plot of the
41 learning rate in the two conditions might be either replaced or accompanied by a scatter plot.

42 We agree and added individual data points for all bar plots.

43

44 (2) the authors controlled for several different factors that could affect the outcome of the
45 study, including expectations, attentional abilities, motivation... however, two basic factors
46 that might be potentially relevant have not been considered. I wonder whether the observed
47 effect of action video games might be different across genders and with age.

48 This is a good point. We addressed this issue in details and added the analysis to the
49 supplements. As a short answer, we did not find any effect of age or gender on the learning
50 parameters.

51 In lines 149-156 of supplementary information, we added a paragraph as below for the initial
52 intervention study:

53 **“Age and gender.** We wonder whether age and gender had an effect on learning
54 parameters and no effect was observed. No difference existed between male and
55 female neither in the orientation learning task (learning rate, $t(50) = 1.07$, $p =$
56 0.29 , Hedge's $g = 0.3$, $BF_{01} = 3.22$; final performance, $t(50) = -0.73$, $p = 0.47$,
57 Hedge's $g = -0.21$, $BF_{01} = 4.34$; initial performance, $t(50) = -1.12$, $p = 0.26$,
58 Hedge's $g = -0.32$, $BF_{01} = 3.12$), nor in the working memory learning task
59 (learning rate, $t(50) = 0.97$, $p = 0.34$, Hedge's $g = 0.27$, $BF_{01} = 3.57$; initial
60 performance, $t(50) = 1.11$, $p = 0.27$, Hedge's $g = 0.31$, $BF_{01} = 3.12$). Age was not
61 correlated with learning rates neither in the orientation learning task ($r = 0.01$, $p =$
62 0.96) nor in the working memory learning task ($r = -0.09$, $p = 0.52$).

63

64 In lines 275-282 of supplementary information, we added a paragraph as below for the
65 replication intervention study:

66 **Age and gender.** We wonder whether age and gender had an effect on learning
67 parameters and no effect was observed. No difference existed between male and
68 female neither in the orientation learning task (learning rate, $t(23) = -0.07$, $p =$
69 0.94 , Hedge's $g = 0.03$, $BF_{01} = 4$; final performance, $t(23) = -0.36$, $p = 0.72$,
70 Hedge's $g = -0.15$, $BF_{01} = 3.84$; initial performance, $t(23) = 0.33$, $p = 0.74$,
71 Hedge's $g = 0.13$, $BF_{01} = 3.84$), nor in the working memory learning task (learning
72 rate, $t(23) = 1.32$, $p = 0.20$, Hedge's $g = 0.53$, $BF_{01} = 1.92$; initial performance,
73 $t(23) = 0.29$, $p = 0.77$, Hedge's $g = 0.12$, $BF_{01} = 3.84$). Age was not correlated with
74 learning rates neither in the orientation learning task ($r = -0.14$, $p = 0.49$) nor in
75 the working memory learning task ($r = -0.21$, $p = 0.31$)."

76

77 (3) The discussion is very clear but a bit succinct, I think it could benefit from some
78 speculation about the mechanisms underlying the effect of active video games on learning
79 abilities, perhaps in the more general framework of neural plasticity.

80 Thank for your suggestion. We know add some speculation of the possible computational and
81 neural mechanisms in lines 271-286 in the main text.

82 In previous work, we have proposed that increased attention control could be one of
83 the mechanisms through which improved learning, whether for perceptual or
84 cognitive learning^{10,29}. In this view, attentional control processes, which encompass
85 cognitive flexibility and working memory, act as a guide to identify and to keep track of
86 task-relevant features, that are relevant to be learnt²³. This is in line with recent
87 computational approaches to learning which also highlight the pivotal role of
88 attention³⁰. Here, however, we did not find evidence that improved learning, as
89 measured by learning rates, was correlated with improved attention control, as
90 measured by the Multiple Object Tracking task. While the role of attentional control
91 on learning rate remains a promising avenue, it could also be the case that a
92 general learning ability exists, that is involved in many unrelated tasks, including
93 orientation and working memory learning tasks³¹. It will be for future studies to
94 further address these important issues.

95 How would these increased learning rates translate in terms of neural plasticity?
96 Action video games encompass remarkably diverse and rapidly changing
97 environments and goals, requiring fast adaptation of mental settings. The
98 orbitofrontal and anterior cingulate cortices have been identified as detecting
99 environmental volatility and controlling learning rate³². Learning rate can also be
100 modulated through a hierarchically higher cognitive control system³³.

101